# Construction and Transition Metal Oxide Loading of Hierarchically Porous Carbon Aerogels

**DOI:** 10.3390/polym12092066

**Published:** 2020-09-11

**Authors:** Jintian Wang, Xinyang Ruan, Jiahao Qiu, Hao Liang, Xingzhong Guo, Hui Yang

**Affiliations:** 1State Key Laboratory of Silicon Materials, School of Materials Science and Engineering, Zhejiang University, Hangzhou 310027, China; 21926060@zju.edu.cn (J.W.); 2009rxy@163.com (X.R.); 3170103280@zju.edu.cn (J.Q.); 3130103649@zju.edu.cn (H.L.); yanghui@zju.edu.cn (H.Y.); 2Pan Asia Microvent Tech (Jiangsu) Coporation & Zhejiang University Micro-nano-porous Materials Joint Research Development Center, Changzhou 213100, China

**Keywords:** carbon aerogel, hierarchically porous material, atmospheric drying, pore formation, transition metal oxide

## Abstract

Hierarchically porous carbon aerogels (CAs) were prepared by organic condensation gelation method combined with atmospheric drying and pore-formation technology, followed by a carbonization process. With as-prepared CAs as substrate, the transition metal oxide nanoparticles loaded CAs composites (MnO_2_/Mn_2_O_3_@CA and Ni/NiO@CA) were achieved by means of liquid etching method combined with heat treatment, respectively. The catalyst, pore-forming agent and etching have important roles on the apparent density and pore structure of CAs. The hydrochloric acid (catalyst) significantly accelerates the gelation process and influences the size and distribution of macropores, whereas the addition of PEG2000 (pore-forming agent) and the etching of liquid solution leads to the formation of mesopore structure in CAs. Appropriate amounts of hydrochloric acid and PEG2000 allow the formation of hierarchically porous CAs with a BET surface area of 482.9 m^2^·g^−1^ and a macropore size of 11.3 μm. After etching and loading, the framework of CAs is etched to become a mesoporous structure, and the transition metal oxide nanoparticles can be uniformly loaded in CAs. These resultant composites have promising application in super capacitor, electrocatalysis, batteries and other fields.

## 1. Introduction

Carbon aerogel (CA), as a derivative of organic aerogel, is usually based on the polycondensation of resorcinol (*R*) and formaldehyde (*F*) [1]. It is a kind of porous amorphous carbon nano-material with a 3D interconnected structure, high porosity (>80%), large specific surface area (usually 400~1100 m^2^·g^−1^ before activation) and a particle diameter of around 3~20 nm [2]. CA is unique among other aerogels because of its electrical conductivity, corrosion resistance and chemical stability [3,4], and has broad prospects in water purification [5,6], drug delivery system [7], heat insulation [8], super capacitor [9,10,11], carrier for catalyst [12,13], batteries [14] and other fields [15,16,17].

Compared with CA with a single pore size, hierarchical porous CA has more unique characteristics and properties derived from the micro–meso–macropore structure. The micropores and mesopores can provide higher surface area and pore volume to facilitate the easy access of electrolyte into the whole system [10,18,19,20], the exposable chemical active sites can improve reaction activity and reaction rate [12,21], and the macropores can increase the porosity and decrease the density of the materials [22,23]. Moreover, the interconnected structure can also be used as transport channels to reduce path lengths for ion diffusion [11]. However, in the present CAs, the large volume shrinkage during the polymerization and drying process of a RF polymer inevitably occurs. Even if the micropore structure of gels could be maintained, the large pore structure in carbon aerogel disappears, which leads to a single pore size distribution and limits its application.

The pore structure of carbon aerogel is closely related to the sol-gel process [24]. Recently, there has been many attempts to regulate the pore structure by organic polymers [25]. Among the approaches, the pore-forming process is efficient approach to create the pores in the CAs [26]. In addition, supercritical/freeze drying are required to remove solvent to preserve the pore structure [27]. As is well known, supercritical/freeze drying is still not a cost-effective drying method due to the high requirements of equipment, complex operation, long experimental period, low output and high cost [28]. Many researchers have shifted their attention to the atmospheric drying method on the basis of retaining the three-dimensional porous network of the wet gel.

Herein, we report the preparation of hierarchical porous CAs by sol-gel combined with atmospheric drying and pore-formation technology. The effects of catalyst on the pore structure of carbon aerogel were investigated in detail. Two transition metal oxides were loaded on hierarchically porous CAs by a liquid etching process and heat treatment. More importantly, these composite materials can incorporate a secondary phase to improve the electrochemical activities of CAs, and the supporting structure stabilizes active components, which will extend the practical applications in energy storage for CAs.

## 2. Results and Discussion

### 2.1. Controllable Preparation of Carbon Aerogels

The synthesis strategy of transition metal oxides loaded CAs is schematically illustrated in Figure 1, and the typical SEM images are also provided to more intuitively demonstrate the synthesis process. Initial red and porous organic aerogels are simply synthesized via a sol-gol process accompanied by pore-formation technology, and the xerogel is mainly composed of pore-forming agent (PEG) and resorcinol-formaldehyde reaction product. Hierarchically porous carbon aerogels (CAs) are carbonized in argon atmosphere, and accordingly, the as-synthesized RF organic aerogels are in situ transformed into carbon aerogels with pore structure. Then, the transition metal oxide nanoparticles are loaded on the skeletons of CAs by etching and immersing, which results from high specific surface area of carbon aerogels. As a result, these composite materials have uniform and controllable morphology.

In the preparation of organic aerogel, the use of catalyst has a significant effect on the organic addition polycondensation, thus controlling the appearance and microstructure of carbon aerogel. Hydrochloric acid is usually used as catalyst for the preparation of resorcinol-formaldehyde organic aerogel, and the acid environment can obviously promote the formation of RF organic gel. The effect of hydrochloric acid on the microstructure of carbon aerogel was investigated. With the increase of the amount of hydrochloric acid, the gelation time is shortened gradually, and the sample without hydrochloric acid has a gelation time as high as more than 4 h. At 60 °C, the sample with 0.5 mL of hydrochloric acid can gelate within 1 h. A supplementary experiment showed that if the same volume of 0.1 M hydrochloric acid was completely replaced by water, the solution will completely gelate in 15 min, accompanied by a severe exothermic phenomenon and a large number of pores on the surface of the xerogels.

The wet gels appear to be pale orange, and there is almost no volume contraction after atmospheric drying. The monolithic RF organic aerogel samples have strong strength, and the color changes to red with a dark oxidation zone on the surface. After carbonization, the samples does not shrink obviously, and the final carbon aerogel has a monolithic shape, high strength and small density. The volume density is 200~400 mg·cm^−3^, which is much lower than other common aerogels.

To study the graphitization of CAs with different amounts of hydrochloric acid (0.1 M), Raman spectra are recorded, as exhibited in Figure 2. All of the CAs show similar spectra including D and G band at around 1340 and 1580 cm^−1^, respectively. Generally, the D band is assigned to the disorder or defect introduced during preparation, while the G band originates from E_2g_ symmetry, which is associated with the degree of graphitization of carbon materials. The intensity ratio of D band and G band (*I_D_/I_G_*) of each sample is at around 1, indicating that the as-prepared carbon aerogels are amorphous carbon with defects and graphitization in some areas. The defects and irregularities promotes the production of micropores [5], while the ordered graphitic structure can improve the electronic conductivity of CAs. [29]

All the morphology of carbon aerogels prepared by different hydrochloric acid amounts are similar. Figure 3 shows the SEM of carbon aerogels prepared by adding 0.1 and 0.5 mL hydrochloric acid, respectively. The results show that the carbon aerogel has a typical 3D cocontinuous skeletons, which are formed by the connection of micron spherical carbon particles. Rich macropores with a large size are formed between the skeletons. With the increase of the amount of hydrochloric acid, the average size of carbon spheres decreases gradually from 8–9 to 4–5 μm, the uniformity is significantly improved, and the skeletons are gradually refined. When the amount of 0.1 M hydrochloric acid exceeds 0.5 mL, the size of carbon spheres tends to be stable with a distribution of 4–5 μm, but the uniformity of carbon spheres decreases and the surface roughness increases obviously, with the appearance of the ellipsoidal particles. These findings are in an agreement that the an appropriate amount of hydrochloric acid accelerates the reaction of this system, resulting in the rise in porosity with ungrown carbon spheres, but excess amount of hydrochloric acid will reduce the uniformity of carbon spheres.

Figure 4 shows the adsorption–desorption isotherms of carbon aerogels with different HCl solution amounts. According to the IUPAC classification, all samples exhibit the isotherm of type I without obvious hysteresis loop, which indicates that there exist only micropores in CAs, almost no mesopores. Additionally, the adsorption capacity increases rapidly in the low pressure region, indicating that there are abundant micropores in the samples [30]. Combined with SEM images, these pores are formed by the accumulation of carbon spheres. Figure 4b shows the pore size distribution of the sample added 0.5 mL hydrochloric acid, which shows that there are abundant micropores in the sample, without mesopores. As shown in Figure 4c, the BET-specific surface area of carbon aerogels increases at first and then decreases with the increase of hydrochloric acid amount. When the amount of hydrochloric acid is 0.5 mL, the specific surface area reaches the highest, which is 645.9 m^2^·g^−1^. Excessive addition of hydrochloric acid will decrease specific surface area of carbon aerogels.

The TG/DTG of organic aerogel in N_2_ atmosphere is shown in Figure 5. With the increase of temperature, the weight loss of RF aerogel occurs continuously, which can be divided into five stages. At 50–140 °C, the weight loss rate is 3.37%, which is contributed by the volatilization of a little underdried solvent and adsorbed water in the pores. At 140–270 °C, the weight loss rate is 3.31%, which is attributed to the further cross-linking curing of phenolic resin and removal of end groups such as methylol (–CH_2_OH). The pyrolysis products of this stage are mainly water (H_2_O), methanol (CH_3_OH) and other small molecules, so the weight loss rate is low. At 270–450 °C, the weight loss rate is 10.15%, which is due to the thermal decomposition process of ortho-ortho and ortho-para methylene. At 450~850 °C, the weight loss rate is 23.75%, which is caused by the decomposition of para-para-methylene and dehydro-carbonization stage and the weight loss rate is high. At 850–900 °C, the mass of sample is stable and almost no weight loss occurs. The total weight loss is about 40.6%, which matches the proportion of non-carbon elements in phenolic resin, indicating that the carbonization reaction is complete.

According to the TG/DTG results of RF aerogel, the highest carbonization temperature is designed to be 900 °C, which ensures that the organic aerogel can be completely carbonized into pure carbon materials, and the xerogel monolith will not shrink or break.

### 2.2. Construction of Hierarchically Porous Carbon Aerogels

In the process of preparing carbon aerogel with hydrochloric acid as catalyst, polyethylene glycol with molecular weight of 200, 400, 800 and 2000 are added as pore-forming agent to explore its effect on the pore structure of carbon aerogel, respectively. When PEG200 is used as pore-forming agent, the amount varies from 0.1 to 0.8 g, the sample collapses on one side after gelation, and occurs obvious delamination phenomenon after drying. It can be seen that the uniformity is poor, and the particle size does not have obvious regularity. When PEG400 and PEG800 are used as pore-forming agents, the microstructure of carbon aerogels are similar to that of PEG200, which also have poor uniformity. When PEG2000 is used as pore forming agent, the gelation time of each sample is within 1 h, and there is no shrinkage after drying. Surface morphologies of carbon aerogels prepared with varied PEG2000 amounts have been observed and recorded by SEM, as shown in Figure 6. With the increase of the addition of PEG2000, the morphologies vary a little, but the particle size of carbon spheres increases significantly. Further increase in PEG2000 will lead to poor uniformity or aggregation of carbon spheres.

The N_2_ adsorption–desorption isotherms of carbon aerogels prepared from different PEG2000 amounts are shown in Figure 7a. According to the IUPAC classification, all samples are found to exhibit an incomplete adsorption isothem of type IV with a elongated hysteresis loop of the H4 type. In addition, the adsorption capacity increases rapidly in the low pressure region, reaching the platform at 220, 190, 160, 152 and 140 cm^3^·g^−1^, respectively, and keeps the platform stable when the pressure continues to increase. These results indicate that there are a large number of micropores or small mesopores in samples [31]. The inner surface area of samples is significantly higher than its outer surface area. The adsorption capacity is limited by the pore volume. The small pores corresponding to the turning point of the platform are completely filled with adsorbed gas.

The micropores size distribution was calculated by the BJH algorithm. From Figure 7b, it is seen that the addition of PEG2000 has little effect on the pore size distribution of carbon aerogels. The micropore sizes of carbon aerogels prepared with different amounts of PEG2000 are distributed in the range of around 0.6~1.0 and 1.8~2.2 nm. With the increase of PEG2000, the micropore volume decreases obviously, which also leads to the decrease of specific surface area, as shown in Table 1. It can be seen that the addition of PEG2000 increases the micropore size slightly with the appearance of small mesopores, while the total pore volume decreases.

The macropore size distribution was calculated by the mercury intrusion method. Figure 8 shows the macropore size distribution of carbon aerogels prepared with different amounts of PEG2000. When the addition of PEG2000 is 0, 0.2, 0.4, 0.6 and 0.8 g, the median macropore size of the samples is distributed about 6.8, 9.1, 11.3, 11.3 and 12.8 μm, respectively. In particular, CA-13 exhibits super-large pore size distribution at about 21.3 μm. These results show that the addition of PEG2000 can effectively expand the original macropore size of carbon aerogels. When the amount of PEG2000 is less than 0.4 g, the increase of its addition obviously enlarges the pore size; however, when the amount is more than 0.4 g, the macropore size tends to be stable.

Therefore, the addition of PEG2000 as a pore-forming agent in hydrochloric acid catalytic system can expand the pore size of carbon aerogels, the original micropore structure can be expanded to mesopores, and the macropore size also increases significantly, thus forming a kind of hierarchical pore structure (micropore-mesopore-macropore). In this process, the specific surface area decreases, but can still be higher than 400 m^2^·g^−1^.

Polyethylene glycol (PEG) is a kind of polymer formed by the polymerization of ethylene oxide. It is a kind of water-soluble polymer with crystallinity and thermoplasticity. Its degree of polymerization can be changed in a large range. PEG is usually viscous liquid or wax like solid. The concentration of active end groups of this kind of resin is low, and it has certain water solubility due to the formation of hydrogen bond.

As is well known, the specific area is mainly produced by the porous structure. Meanwhile, the mesopores in CA matrix are mainly distributed between the particles, and the micropores mainly exist in the particles. However, the microstructural collapse and shrinkage of CA during carbonization lead to the disappear of mesopores and macropores. Therefore, the micropores is dominant in CAs [14]. In the system of hydrochloric acid, the added PEG2000 acts as a soft template and structure-directing agent in the reaction [25]. The condensation of resorcinol and formaldehyde small organic molecules around the PEG chain expands. Under suitable conditions, the structure with PEG2000 template and RF gel as support is formed. Due to the thermosetting of RF gel and the three-dimensional network structure strengthened by PEG2000, the volume shrinkage decreases. Additionally, RF gel also converts to carbon after carbonization, resulting in more abundant pores in aerogel and retaining the carbon skeletons formed after pyrolysis of organic gels.

### 2.3. Transition Metal Oxides Loading of Hierarchically Porous Carbon Aerogels

#### 2.3.1. Nano Manganese Oxide Loading of Carbon Aerogels

X-ray photoelectron spectroscopy (XPS) is performed to investigate the chemical state of the elements in MnO_x_@CA, as shown in Figure 9a,b. The wide survey spectrum confirms the existence of Mn, O and C elements, and the Mn loading content is 2.89%. From the high-resolution Mn 2p XPS spectrum of MnO_x_@CA, it is obviously observed that the Mn 2p doublet is corresponding to 2p_1/2_ and 2p_3/2_ splitting peak. According to the flitting results, the Mn 2p_3/2_ spectra can be dissolved into two peaks at 643.0 and 641.5 eV, respectively. This demonstrates two different kinds of chemical states for Mn in MnO_x_@CA. The main peak (centered at about 643.0 eV) is ascribed to Mn^4+^, while the peaks at binding energies of 641.5 eV may be attributed to Mn^3+^. In addition, two corresponding peaks at 654.5 and 653.1 eV are also observed, which are assigned to Mn 2p_1/2_ spectra of Mn^4+^ and Mn^3+^. Moreover, the Mn 2p_1/2_ and 2p_3/2_ are located at 654.3 and 642.6 eV, respectively, and the splitting energy level (Δ*E*_b_) is 11.7 eV, which is consistent with that previously reported [32]. The intensity and area of the peak clearly demonstrate that MnO_x_ is MnO_2_ with a small amount of Mn_2_O_3_.

The phase composition of MnO_2_/Mn_2_O_3_@CA is evaluated by XRD analysis. As shown in Figure 9c, there has been weak characteristic peak of manganese oxide, which may be mainly covered by carbon substrate due to the small load. However, it can still be distinguished into two different phases. Two diffraction peaks at 36.8 and 65.7° correspond to (006) and (119) planes of MnO. Three peaks appearing at 33.2, 43.3 and 60.5° correspond to (−102), (−221) and (−313) panes of Mn_2_O_3_. It can be confirmed that the as-prepared product is with no other impurities.

To further investigate the morphology and microstructure of MnO_2_/Mn_2_O_3_@CA, scanning electron microscopy (SEM) is carried out. As shown in Figure 10, the carbon spheres on the carbon aerogel skeletons are etched to form a large number of pores, shaping into a nano-flower structure, which provides more potential active sites and accommodation space for loading manganese oxide nanoparticles. Mn and O are uniformly distributed on the surface and inside the pores of carbon spheres, which can be estimated by EDS analysis, but the loading of Mn element is not high. Combined with the results of XPS and XRD analysis, the uniform loading of MnO_2_/Mn_2_O_3_ nanoparticles in hierarchically porous carbon aerogel is successfully achieved by liquid phase synthesis, and MnO_2_/Mn_2_O_3_@CA has uniform and controllable morphology.

Nitrogen adsorption–desorption isotherm of MnO_2_/Mn_2_O_3_@CA sample is shown in Figure 11. Difference from the uncomposite carbon aerogel, the curve of loaded composite belongs to the typical type IV isotherm according to the IUPAC classification. Such isotherm shows an obvious hysteresis loop at the *P*/*P*_0_ between 0.10 and 0.80, owing to the presence of a large amount of mesoporous in the sample, which is consistent with the morphology of SEM. The BET-specific surface area of MnO_2_/Mn_2_O_3_@CA is as high as 505.7 m^2^·g^−1^.

#### 2.3.2. Nano Nickel Oxide Loading of Carbon Aerogels

XPS is also performed to investigate the chemical state of the elements in NiO_x_@CA, as shown in Figure 12a,b. It can be observed from Figure 12a that the survey spectrum reveals the presence of the elements nickel and oxygen in the structure of CA, and the Ni loading content is 1.37%. The high-resolution XPS spectra of NiO_x_@CA corresponding to the binding energies for the Ni 2p is analyzed in Figure 12b. Typical prominent doublet peaks of the sample are located at approximately 872.58 and 853.78 eV, with one satellite peak at 880.03 and 861.03 eV, corresponding to the Ni 2p_1/2_ and Ni 2p_3/2_ main level binding energies of Ni, respectively. Additionally, the Ni 2p level splitting energy (Δ*E*_b_) is 18.8 eV, which is consistent with that previously reported [30]. The Ni 2p peaks in NiO_x_@CA are observed at 873.43 and 855.48 eV, which may be attributed to the presence of Ni^2+^, and other two peaks may correspond to Ni^0^ (872.13 and 853.68 eV).

The XRD pattern of the obtained composite is shown in Figure 12c. The peaks at 37.12°, 43.12°, 62.71°, 75.35° and 79.29° can be assigned to (111), (002), (022), (311) and (222) of cubic NiO, respectively, and the peaks at 44.32, 51.67 and 76.13°can be assigned to (111), (002) and (022) of Ni with the same cubic structure as NiO. According to the Bragg equation, the lattice constants of cubic NiO and cubic Ni are 4.180 and 3.540 Å, respectively, which are consistent with that previously reported [33,34]. All of the sharp diffraction and defined peaks indicate that the sample is highly crystallized, and no other obvious impurity peaks are detected.

The morphology and structure of the obtained Ni/NiO@CA are characterized by scanning electron microscopy (SEM). Figure 13 shows that the carbon spheres are etched to form a large number of pores after loading. Unlike the previous Mn loading, most of these pores are mesopores that are smaller than 50 nm, but they can also provide many potential active sites and accommodation space for loading nickel and nickel oxide nanoparticles. After loading, the skeletons of Ni/NiO@CA still maintain a complete carbon spherical structure, with no obvious structural damage or collapse. In addition, the elemental mapping images of Ni and O (Figure 13) verify the homogeneous distribution of nickel and nickel oxide on the surface and inside the pores of carbon aerogel, but the loading of Ni element is not high. Combined with the results of XPS and XRD analysis, the uniform loading of Ni/NiO nanoparticles in hierarchically porous carbon aerogel is successfully achieved by liquid phase synthesis, and the Ni/NiO@CA has uniform and controllable morphology.

To further investigate the porosity of Ni/NiO@CA composite material, the nitrogen adsorption–desorption isotherm measurement was also employed, and the isotherm is shown in Figure 14a. The curve exhibits a mixture of type Ⅰ and IV following the IUPAC classification. The pore size distribution curve which is depicted in Figure 14b also confirms the micro-mesopore structure in the composites.

As a kind of chemical activation method, liquid etching has been widely used in the activation of carbon materials. In this transition metal oxides loading process, the chemical reagents (ammonia solution, NH_4_HCO_3_) react with carbon aerogels, which results in the liquid etching of carbon spheres. The carbon in the carbon spheres will be consumed and developed to be the pores, and the reactive gas (CO_2_, NH_3_) will expand the micropores to form large mesopores. It indicates that liquid etching is an effective approach to construct mesopore structure.

## 3. Experimental Section

### 3.1. Materials

Resorcinol (*R*, CAS 108-46-3, Aladdin, Shanghai, China, 99%), formaldehyde solution (*F*, CAS 50-00-0, Sinopharm Chemical Reagent Co., Ltd., Shanghai, China, 37%~40%), hydrochloric acid (HCl, CAS 7647-01-0, Sinopharm Chemical Reagent Co., Ltd., Shanghai, China, 36~38%), ethanol (CAS 64-17-5, Sinopharm Chemical Reagent Co., Ltd., Shanghai, China, AR), polyethylene glycol (PEG, CAS 25322-68-3, Aladdin, Shanghai, China, AR), nickel nitrate hexahydrate (Ni(NO_3_)_2_·6H_2_O, CAS 13478-00-7, Aladdin, Shanghai, China, AR), manganese chloride (MnCl_2_, CAS 7773-01-5, Aladdin, Shanghai, China, 99%) were used without further purification in this experimental.

### 3.2. Preparation and Loading of Hierarchically Porous Carbon Aerogels

#### 3.2.1. Preparation of Hierarchically Porous Carbon Aerogels

Samples were prepared with the starting compositions listed in Table 2. Firstly, catalyst (0.1 mol/L HCl) and polyethylene glycol (PEG, *M**_w_* = 200, 400, 800 and 2000) were dissolved in deionized water. Then, 2.75 g resorcinol (R) and 3.61 mL formaldehyde solution (F) were added to the solution. After stirring for 1 h, the RF solution was sealed and cured in 60 °C for 24 h. The resulting RF wet gels with dark red in color were immersed in ethanol for 24 h and then dried in 45 °C for 2 days to prepare RF aerogels. The organic aerogels were carbonized in argon at 900 °C for 1 h at a heating rate of 1 °C·min^−1^. Finally, the carbon aerogels (CAs) were obtained.

#### 3.2.2. Loading of Hierarchically Porous Carbon Aerogels

*Synthesis of MnO_2_/Mn_2_O_3_@CA.* 0.25 g carbon aerogel was added to 10 mL 3 M ammonia solution, stirred under 90 °C oil bath for 20 h. Then, the precipitation was added to 10 mL 0.05 M MnCl_2_, stirred under 75 °C oil bath for 24 h, and dried in 60 °C. After that, the sample was heat treated in argon at 500 °C for 3 h at a heating rate of 1 °C/min. Finally, the carbon aerogel loaded with MnO_2_/Mn_2_O_3_ nanoparticles was obtained.

*Synthesis of Ni/NiO@CA.* 0.25 g carbon aerogel and 145.5 mg Ni(NO_3_)_2_·6H_2_O powder were added to 10 mL deionized water and stirred at room temperature for 30 min, then the solution was added 5 mL 0.5 M NH_4_HCO_3_ and stirred for 24 h. After dried in 60 °C, the sample was heat-treated in argon at 600 °C for 2 h at a heating rate of 1 °C·min^−1^. Finally, the carbon aerogel loaded with Ni/NiO nanoparticles was obtained.

The production yield (*γ*) of transition metal oxides in carbon aerogel calculated by the formula: *γ* = molar mass of the Mn (or Ni) loaded/ molar mass of the Mn (or Ni) in the raw material. After calculation, the *γ* of MnO_2_/Mn_2_O_3_ in CAs is about 80.64%, and the *γ* of Ni/NiO in CAs is about 57.88%.

### 3.3. Characterization

The Raman spectra were recorded from 1000 to 2800 cm^−1^ using a micro Raman apparatus (LabRAM HR Evolution, Horiba Jobin Yvon, Paris, France). The microstructures of the synthesized materials were observed by the scanning electron microscope (SEM, SU-8010, Hitachi, Tokyo, Japan) coupled with energy-dispersive X-ray spectroscopy (EDS). The specific surface area was determined by nitrogen adsorption (BET-method, ASAP2460, Micromeritics Instruments Corporation, Norcross, GA, USA), and the macropore size distribution was determined by mercury injection apparatus (AutoPore IV 9510, Micromeritics Instruments Corporation, Norcross, GA, USA). Phase identification of prepared samples was done by X-ray diffractometer (XRD, D8 ADVANCE, Bruker AXS, Karlsruhe, Germany) and X-ray photoelectron spectrometer (XPS, ESCALAB 250Xl, ThermoFisher Scientific, Waltham, MA, USA).

## 4. Conclusions

Hierarchically porous carbon aerogels and their composites were prepared by process optimization, pore structure regulation and nano-loading. When the amount of HCl solution (0.1 M) is 0.5 mL, and the amount of PEG2000 used as pore-formation agent is 0.4 g, micro–meso–macroporous CAs with a BET surface area of 482.9 m^2^·g^−1^ and a macropore size of 11.3 μm can be achieved. With hierarchically porous carbon aerogels as substrates, the transition metal oxide nanoparticles were loaded to prepare MnO_2_/Mn_2_O_3_@CA and Ni/NiO@CA composite materials by means of etching and loading combined with heat treatment. After loading, the framework of carbon aerogels is etched into mesoporous structure with a specific surface area as high as 625.9 m^2^·g^−1^ and the transition metal oxide nanoparticles distribute uniformly in CAs. The construction of hierarchical porous structure, the loading of transition metal oxide nanoparticles and atmospheric drying technology will reduce the preparation cost and widen the application range of carbon aerogels.

## Figures and Tables

**Figure 1 polymers-12-02066-f001:**
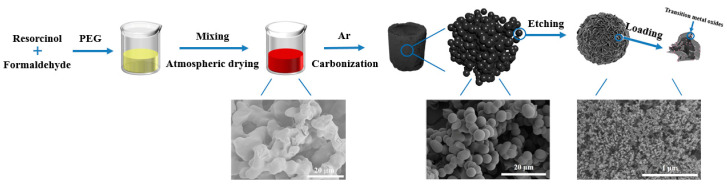
Schematic illustration of the synthesis process for transition metal oxides@CA.

**Figure 2 polymers-12-02066-f002:**
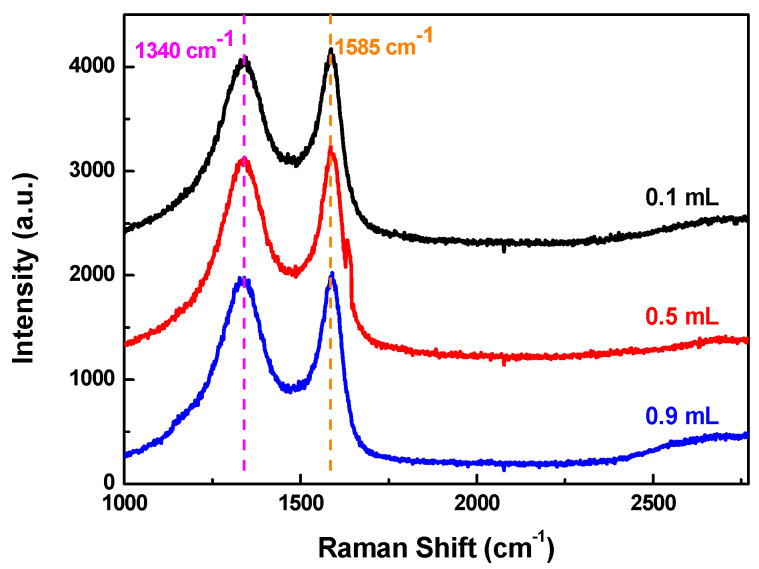
Raman spectrum of CAs prepared with different amounts of HCl solution (0.1 M).

**Figure 3 polymers-12-02066-f003:**
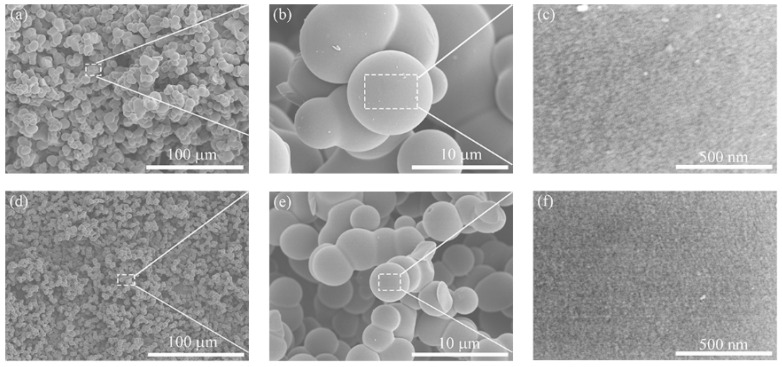
SEM images of carbon aerogels prepared with different amounts of 0.1 M HCl solution: (**a**–**c**) 0.1 mL; (**d**–**f**) 0.5 mL.

**Figure 4 polymers-12-02066-f004:**
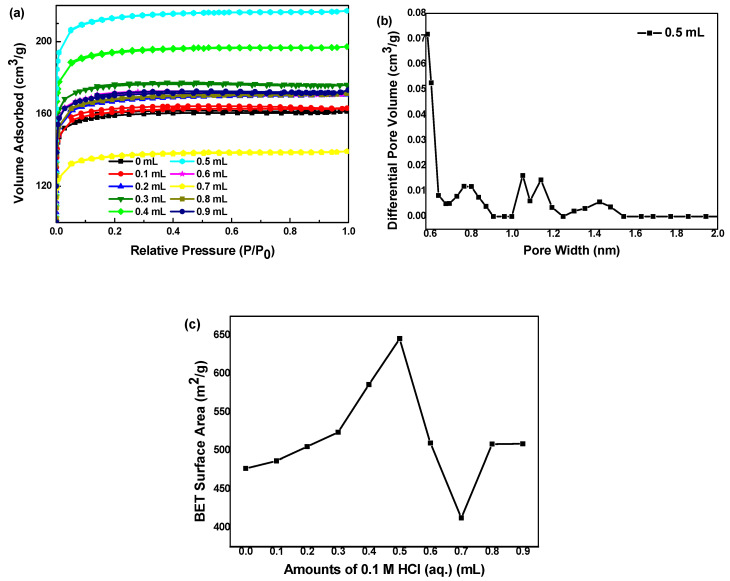
BET results of carbon aerogels prepared with different amounts of 0.1 M HCl solution: (**a**) N_2_ adsorption and desorption curves; (**b**) The pore diameter distribution curve calculated by BJH model of the sample prepared with 0.5 mL HCl solution; (**c**) Relationship between BET-specific surface area and HCl solution amount.

**Figure 5 polymers-12-02066-f005:**
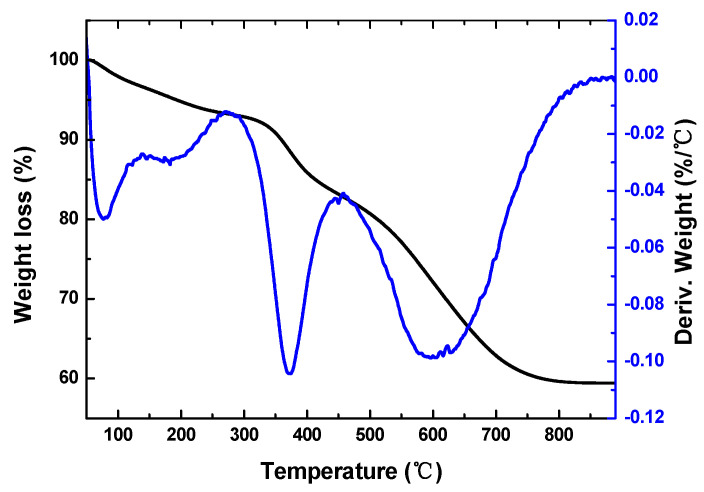
TG/DTG curves of resorcinol-formaldehyde (RF) organic aerogel.

**Figure 6 polymers-12-02066-f006:**
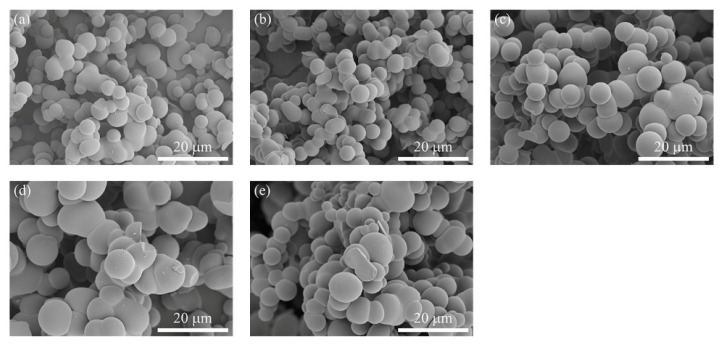
SEM images of carbon aerogels prepared with different PEG2000 amounts: (**a**) 0 g; (**b**) 0.2 g; (**c**) 0.4 g; (**d**) 0.6 g; (**e**) 0.8 g.

**Figure 7 polymers-12-02066-f007:**
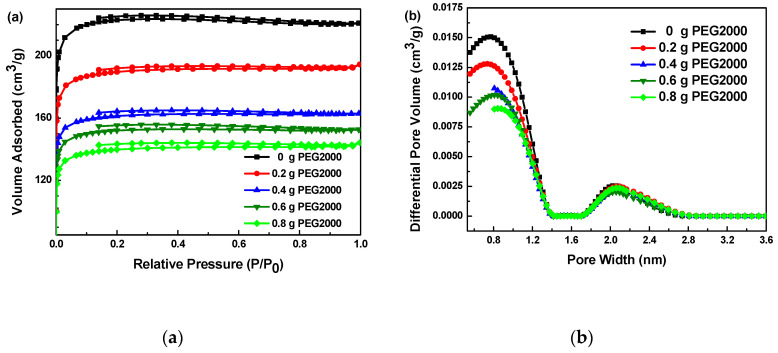
BET results of carbon aerogels prepared with different amounts of PEG2000: (**a**) N_2_ adsorption and desorption curves; (**b**) The pore diameter distribution curves calculated by BJH model.

**Figure 8 polymers-12-02066-f008:**
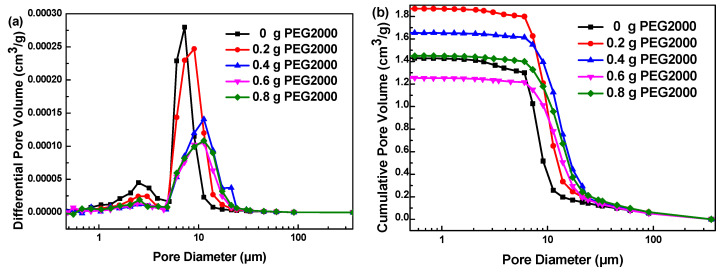
Macropore size distribution of carbon aerogels prepared with different amounts of PEG2000: (**a**) differential pore volume curve; (**b**) cumulative pore volume curve.

**Figure 9 polymers-12-02066-f009:**
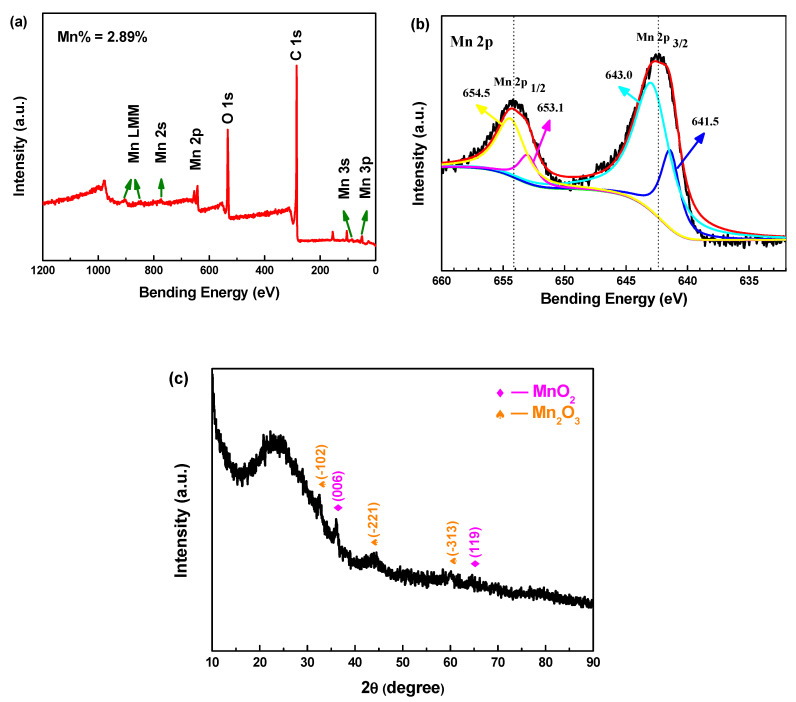
(**a**) Wide survey XPS spectrum, (**b**) high-resolution Mn 2p XPS spectrum; and (**c**) XRD spectra of MnO_2_/Mn_2_O_3_@CA composite material.

**Figure 10 polymers-12-02066-f010:**
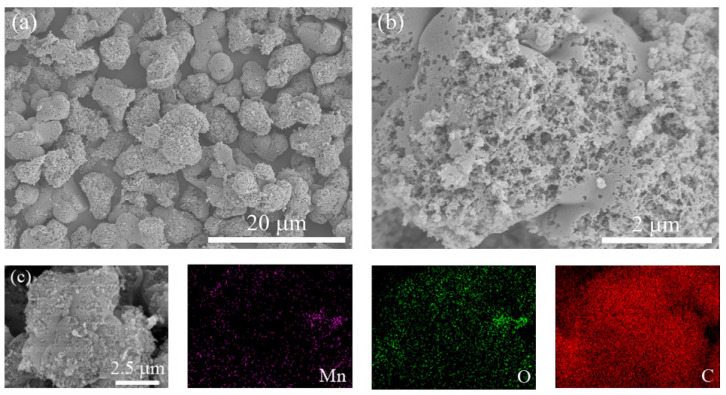
(**a**,**b**) SEM images; (**c**) EDS results of MnO_2_/Mn_2_O_3_@CA composite material.

**Figure 11 polymers-12-02066-f011:**
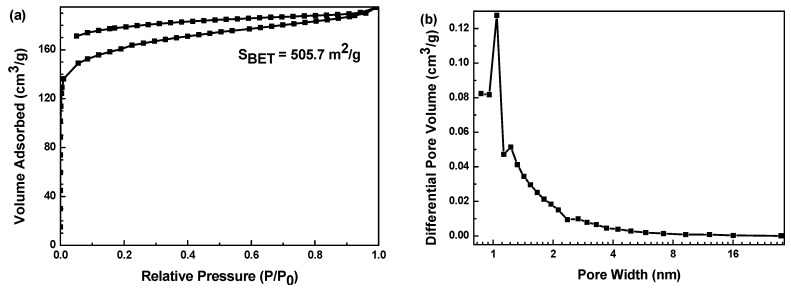
(**a**) N_2_ adsorption–desorption isotherm; (**b**) pore diameter distribution of MnO_2_/Mn_2_O_3_@CA composite material.

**Figure 12 polymers-12-02066-f012:**
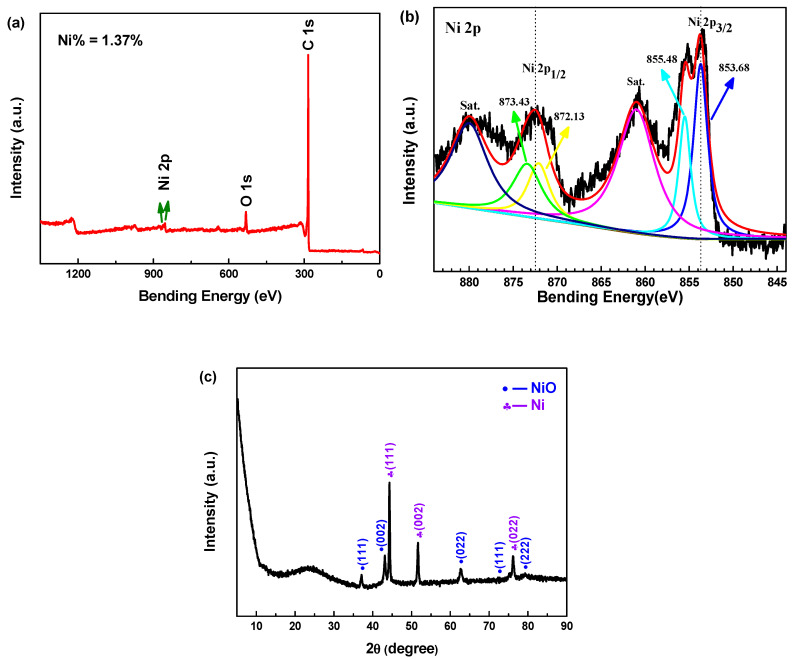
(**a**) Wide survey XPS spectrum; (**b**) high-resolution Ni 2p XPS spectrum; (**c**) XRD spectra of Ni/NiO@CA composite material.

**Figure 13 polymers-12-02066-f013:**
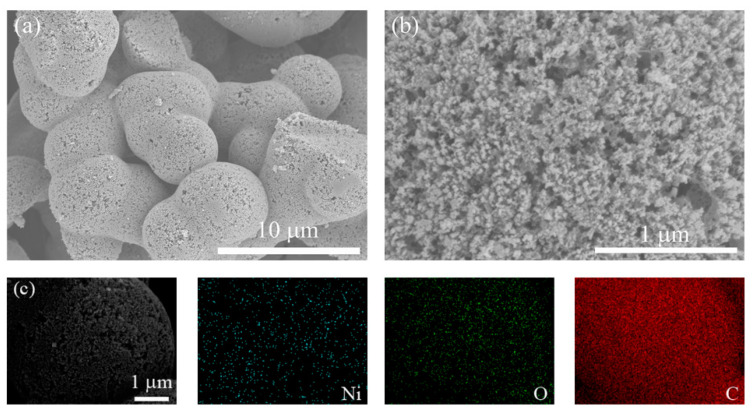
(**a**,**b**) SEM images; (**c**) EDS results of Ni/NiO@CA composite material.

**Figure 14 polymers-12-02066-f014:**
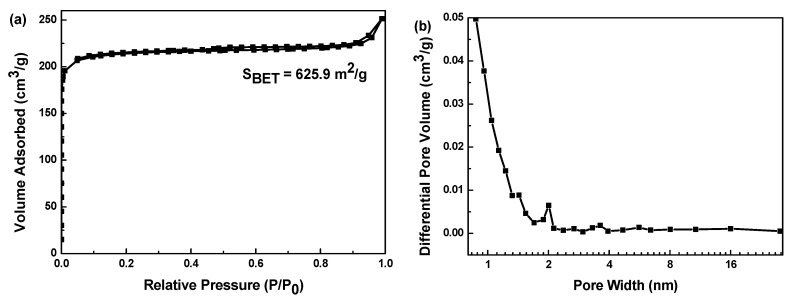
(**a**) N_2_ adsorption–desorption isotherm; (**b**) pore diameter distribution of Ni/NiO@CA composite material.

**Table 1 polymers-12-02066-t001:** Specific surface areas of carbon aerogels prepared with different amounts of PEG2000.

PEG2000/g	0	0.2	0.4	0.6	0.8
BET-specific surface area/m^2^·g^−1^	665.1	568.8	482.9	455.1	419.2
Pore volume/cm^3^·g^−1^	0.343	0.297	0.252	0.235	0.217
Micropore surface area/m^2^·g^−1^	605.1	509.2	434.8	410.1	372.6
Micropore volume/cm^3^·g^−1^	0.318	0.267	0.228	0.215	0.196

**Table 2 polymers-12-02066-t002:** Starting compositions of samples.

Sample	HCl/mL	PEG/g	Water/mL	R/g	F/mL
CA-1	0	0	7.45	2.75	3.61
CA-2	0.1	0	7.45	2.75	3.61
CA-3	0.2	0	7.35	2.75	3.61
CA-4	0.3	0	7.25	2.75	3.61
CA-5	0.4	0	7.15	2.75	3.61
CA-6	0.5	0	7.05	2.75	3.61
CA-7	0.6	0	6.95	2.75	3.61
CA-8	0.7	0	6.85	2.75	3.61
CA-9	0.8	0	6.75	2.75	3.61
CA-10	0.9	0	6.65	2.75	3.61
CA-11	1.0	0	6.55	2.75	3.61
CA-12	0.5	0.2	7.55	2.75	3.61
CA-13	0.5	0.4	7.55	2.75	3.61
CA-14	0.5	0.6	7.55	2.75	3.61
CA-15	0.5	0.8	7.55	2.75	3.61

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
