# Peer review of "Construction and Transition Metal Oxide Loading of Hierarchically Porous Carbon Aerogels"

_polymers, 2020, doi:10.3390/polym12092066_

Round 1

Reviewer 1 Report

The Manuscript entitled Construction and Transition Metal Oxides Loading of 2 Hierarchically Porous Carbon Aerogels,

922127 - polymers.

Recommendation: Publish after minor revisions noted.

Comments:

The manuscript abounds fruitful discussion. I recommend publication of the manuscript after considering some minor remarks.

Which are the CAS Numbers of used chemicals ?

They were used without further purification, or they are purified ? If chemicals were purified, how it was done ?

The English language, typos, grammatical errors throughout the Manuscript should be carefully revised.

The experiments seem to be carried out carefully and thus the data are reliable. The treatment of data is correct and the obtained results are new and interesting. This paper could be suitable for publication in this journal POLYMERS while some improvements as mentioned in the letter above have been made.

Author Response

Response to Reviewer 1 Comments

Point 1: Which are the CAS Numbers of used chemicals ?

Response 1: Thank you for your reminding. CAS Numbers are added.

Resorcinol (R, CAS 108-46-3, Aladdin, Shanghai, China, 99%), formaldehyde solution (F, CAS 50-00-0, Sinopharm Chemical Reagent Co., Ltd., Shanghai, China, 37%~40%), hydrochloric acid (HCl, CAS 7647-01-0, Sinopharm Chemical Reagent Co., Ltd., Shanghai, China, 36~38%), ethanol (CAS 64-17-5, Sinopharm Chemical Reagent Co., Ltd., Shanghai, China, AR), polyethylene glycol (PEG, CAS 25322-68-3, Aladdin, Shanghai, China, AR), nickel nitrate hexahydrate (Ni(NO3)2·6H2O, CAS 13478-00-7, Aladdin, Shanghai, China, AR), manganese chloride (MnCl2, CAS 7773-01-5, Aladdin, Shanghai, China, 99%) were used in the experiment without any further purification.

Point 2: They were used without further purification, or they are purified ? If chemicals were purified, how it was done ?

Response 2: Thank you for your comments. All chemicals were used without any further purification in the experiment.

In 3.1 Materials, ...... were used without further purification in this experimental.

Point 3: The English language, typos, grammatical errors throughout the Manuscript should be carefully revised.

Response 3: Thank you for your suggestion. The English in this manuscript has been modified.

Reviewer 2 Report

In this manuscript, the authors reported MnO2/Mn2O3@CA and Ni/NiO@CA composite materials employing etching and loading combined with heat treatment. As-prepared metal oxide doped carbon aerogels show a specific surface area as high as 625.9 m2 ·g-1.   I agree that the proposed synthesis methods by using the organic condensation gelation method combined with atmospheric drying and pore-formation technology, followed by the carbonization process are attractive. Besides, the resulting performance of materials is outstanding. Therefore, I recommend the current manuscript for publication in the journal of Polymers the following points are addressed:  

  • The authors prepared the hierarchically porous carbon aerogels only at 900 oC (activating step). Why they did not try different temperatures that were applied for the optimizing process?
  • What is the MnO2/Mn2O and Ni/NiO at carbon aerogels? Survey analysis of XPS should be included and metal loading content will be added to the manuscript
  • The production yield of metal oxide loaded carbon aerogels should be calculated and included in the manuscript
  • The introduction part should be re-written to upgrade the importance of this work, and the discussion part should be doing more explanation for the discussion in depth. But I found there is more description of the experimental phenomenon with little discussion for the possible reasons.

  • In the result and discussion part, the comparison of present results with the results of other similar investigations in literature should be discussed in detail and the novelty and the critical improvements in this paper should be defined clearly. This is needed to place this work in perspective with other works in the field and provide more credibility for the present results. Some important and very closely related literature are A) Palapati, N. K. R., et al. "Enhancing the electronic conductivity of Lignin-sourced, sub-micron carbon particles." 2015 IEEE Nanotechnology Materials and Devices Conference (NMDC). IEEE, 2015. B) Ashourirad, Babak, et al. "Rapid transformation of heterocyclic building blocks into nanoporous carbons for high-performance supercapacitors." RSC advances22 (2018): 12300-12309. C) Altinci, Osman Cem, and Muslum Demir. "Beyond Conventional Activating Methods, A Green Approach for the Synthesis of Bio-Carbon and its supercapacitor electrode performance." Energy & Fuels (2020).

Author Response

Response to Reviewer 2 Comments

Point 1: The authors prepared the hierarchically porous carbon aerogels only at 900 ℃ (activating step). Why they did not try different temperatures that were applied for the optimizing process?

 Response 1: Thank you for your comments. We have explored the effect of carbonization temperature on the structure of carbon aerogels. The results show that when the temperature is lower than 900 oC, the RF organic aerogels can not be completely carbonized into carbon aerogels. When the temperature is a little higher than 900 oC, the microstructure of the sample is similar to that prepared with 900 oC. However, when the temperature is much higher than 900 oC, the sample is broken into small pieces with extremely high hardness after carbonization. Therefore, in this experimental, the carbonization temperature is designed to be 900 oC.

Point 2: What is the MnO2/Mn2O3 and Ni/NiO at carbon aerogels? Survey analysis of XPS should be included and metal loading content will be added to the manuscript.

Response 2: Thank you for your comments. XPS analysis has been used to reveal MnO2/Mn2O3 or Ni/NiO loading content in the transition metal oxides@CA composite. They are as follows.

Nano manganese oxide loading of carbon aerogels

X-ray photoelectron spectroscopy (XPS) is performed to investigate the chemical state of the elements in MnOx@CA, as shown in Figures 9(a) and 9(b). The wide survey spectrum confirms the existence of Mn, O and C elements, and the Mn loading content is 2.89%. From the high-resolution Mn 2p XPS spectrum of MnOx@CA, it is obviously observed that the Mn 2p doublet is corresponding to 2p1/2 and 2p3/2 splitting peak. According to the flitting results, the Mn 2p3/2 spectra can be dissolved into two peaks at 643.0 and 641.5 eV, respectively. This demonstrates two different kinds of chemical states for Mn in MnOx@CA. The main peak (centered at about 643.0 eV) is ascribed to Mn4+, while the peaks at binding energies of 641.5 eV may be attributed to Mn3+. In addition, two corresponding peaks at 654.5 and 653.1 eV are also observed, which are assigned to Mn 2p1/2 spectra of Mn4+ and Mn3+. Moreover, the Mn 2p1/2 and 2p3/2 are located at 654.3 and 642.6 eV, respectively, and the splitting energy level (â–³Eb) is 11.7 eV, which is consistent with that previously reported[32]. The intensity and area of the peak clearly demonstrate that MnOx is MnO2 with a small amount of Mn2O3.

Figure 9. (a)Wide survey XPS spectrum, (b) high-resolution Mn 2p XPS spectrum of MnO2/Mn2O3@CA composite material

Ni/NiO@CA is Ni and NiO loaded on the carbon aerogels.

XPS is also performed to investigate the chemical state of the elements in NiOx@CA, as shown in Figures 12(a) and 12(b). It can be observed from Figure 12(a) that the survey spectrum reveals the presence of the elements nickel and oxygen in the structure of CA, and the Ni loading content is 1.37%. The high-resolution XPS spectra of NiOx@CA corresponding to the binding energies for the Ni 2p is analyzed in Figure 12(b). Typical prominent doublet peaks of the sample are located at approximately 872.58 and 853.78 eV, with one satellite peak at 880.03 and 861.03 eV, corresponding to the Ni 2p1/2 and Ni 2p3/2 main level binding energies of Ni, respectively. And the Ni 2p level splitting energy (â–³Eb) is 18.8 eV, which is consistent with that previously reported. The Ni 2p peaks in NiOx@CA are observed at 873.43 and 855.48 eV, which may be attributed to the presence of Ni2+, and other two peaks may correspond to Ni0 (872.13 and 853.68 eV).

Figure 12. (a)Wide survey XPS spectrum, (b) high-resolution Ni 2p XPS spectrum of Ni/NiO@CA composite material

Point 3: The production yield of metal oxide loaded carbon aerogels should be calculated and included in the manuscript

Response 3: Thank you for your comments. The production yield (γ ) of transition metal oxides in carbon aerogel is investigated as follows.

The production yield (γ ) of transition metal oxides in carbon aerogel calculated by the formula: γ = molar mass of the Mn(or Ni) loaded/ molar mass of the Mn(or Ni) in the raw material. After calculation, the γ of MnO2/Mn2O3 in CAs is about 80.64%, and the γ of Ni/NiO in CAs is about 57.88%.

The above sentence is added in the experimental section in the manuscript.

Point 4: The introduction part should be re-written to upgrade the importance of this work, and the discussion part should be doing more explanation for the discussion in depth. But I found there is more description of the experimental phenomenon with little discussion for the possible reasons.

Response 4: Thank you for your comments. The introduction in this manuscript is modified. As for the discussion part, we try to increase some discussion on the formation of mesopore structure in the composites.

Point 5: In the result and discussion part, the comparison of present results with the results of other similar investigations in literature should be discussed in detail and the novelty and the critical improvements in this paper should be defined clearly. This is needed to place this work in perspective with other works in the field and provide more credibility for the present results. Some important and very closely related literature are A) Palapati, N. K. R., et al. "Enhancing the electronic conductivity of Lignin-sourced, sub-micron carbon particles." 2015 IEEE Nanotechnology Materials and Devices Conference (NMDC). IEEE, 2015. B) Ashourirad, Babak, et al. "Rapid transformation of heterocyclic building blocks into nanoporous carbons for high-performance supercapacitors." RSC advances22 (2018): 12300-12309. C) Altinci, Osman Cem, and Muslum Demir. "Beyond Conventional Activating Methods, A Green Approach for the Synthesis of Bio-Carbon and its supercapacitor electrode performance." Energy & Fuels (2020).

Response 5: Thank you for your comments. I have read the literature you recommend carefully and compared this work with other works in the field.

Round 2

Reviewer 2 Report

I suggest the acceptance of revised manuscript